# Maternal Vitamin D and Newborn Telomere Length

**DOI:** 10.3390/nu13062012

**Published:** 2021-06-11

**Authors:** Lisa Daneels, Dries S. Martens, Soumia Arredouani, Jaak Billen, Gudrun Koppen, Roland Devlieger, Tim S. Nawrot, Manosij Ghosh, Lode Godderis, Sara Pauwels

**Affiliations:** 1Centre Environment & Health, Department of Public Health and Primary Care, KU Leuven, 3000 Leuven, Belgium; daneels.lisa@gmail.com (L.D.); tim.nawrot@uhasselt.be (T.S.N.); manosij.ghosh@kuleuven.be (M.G.); lode.godderis@kuleuven.be (L.G.); 2Centre for Environmental Sciences, Hasselt University, 3500 Hasselt, Belgium; dries.martens@uhasselt.be; 3Department of Laboratory Medicine, Leuven University Hospitals, 3000 Leuven, Belgium; soumia.arredouani@uzleuven.be (S.A.); jaak.billen@uzleuven.be (J.B.); 4VITO-Health, Flemish Institute of Technological Research (VITO), 2400 Mol, Belgium; gudrun.koppen@vito.be; 5Department of Development and Regeneration, KU Leuven-University of Leuven, 3000 Leuven, Belgium; roland.devlieger@uzleuven.be; 6Department of Obstetrics and Gynecology, University Hospitals of Leuven, 3000 Leuven, Belgium; 7IDEWE, External Service for Prevention and Protection at Work, 3000 Heverlee, Belgium

**Keywords:** vitamin D, telomere length, pregnancy, newborn

## Abstract

Nutrition is important during pregnancy for offspring health. Gestational vitamin D intake may prevent several adverse outcomes and might have an influence on offspring telomere length (TL). In this study, we want to assess the association between maternal vitamin D intake during pregnancy and newborn TL, as reflected by cord blood TL. We studied mother–child pairs enrolled in the Maternal Nutrition and Offspring’s Epigenome (MANOE) cohort, Leuven, Belgium. To calculate the dietary vitamin D intake, 108 women were asked to keep track of their diet using the seven-day estimated diet record (EDR) method. TL was assessed in 108 cord blood using a quantitative real-time PCR method. In each trimester of pregnancy, maternal serum 25-hydroxyvitamin D (25-OHD) concentration was measured. We observed a positive association (β = 0.009, *p*-value = 0.036) between newborn average relative TL and maternal vitamin D intake (diet + supplement) during the first trimester. In contrast, we found no association between average relative TL of the newborn and mean maternal serum 25-OHD concentrations during pregnancy. To conclude, vitamin D intake (diet + supplements), specifically during the first trimester of pregnancy, is an important factor associated with TL at birth.

## 1. Introduction

Maternal nutrition during pregnancy can influence fetal growth and development and eventually the health of the child [1]. There is a high prevalence of vitamin D deficiency (serum 25-hydroxyvitamin D (25-OHD) < 20 ng/mL)), due to low dietary and supplemental vitamin D intake, low sun exposure, and vitamin D related gene polymorphisms, in 30–60% of the population in Western Europe [2]. In addition, studies have shown a high prevalence of vitamin D deficiency during pregnancy (45% of Belgian pregnant women), which is a significant factor influencing the long-term health of the offspring [3,4]. It is relatively well established that diet has a profound effect on DNA integrity, epigenetic mechanisms (such as DNA methylation), and telomere length. Telomeres are nucleoprotein structures containing TTAGGG repeats at the end of the chromosomes. They are important in maintaining genomic stability and protect chromosomes from end-to-end fusion and degradation. TLs shorten with each cellular division. This process is accelerated with inflammation and oxidative stress. TL is considered a biological marker of ageing and short TL has been associated with age-related diseases [5]. TL at birth represents an individual’s initial setting of TL and predicts later life TL. Martens et al. [6] found that cord blood TL/placenta TL highly correlates with leukocyte TL at the age of 4. Newborn TL is highly variable and environmental factors that influence newborn TL may be important in the contribution of later life TL and the later life TL-related health conditions [7]. Several studies have investigated the importance of in utero exposures in relation to newborn TL, such as maternal pre-pregnancy BMI [5], prenatal secondhand smoke [8], and prenatal stress [9]. However, only a few studies have examined the impact of maternal nutrition on newborn TL. Maternal folate concentration in early pregnancy [10], maternal vitamin C intake [11], and maternal vitamin D concentrations [12] are associated with longer newborn TL. Kim et al. [12] found that newborn leukocyte telomere length was positively correlated with third trimester maternal serum 25-OHD concentrations. However, no study has investigated the association between maternal vitamin D intake and serum levels during each trimester of pregnancy and newborn TL.

In this birth cohort study, we aimed to investigate the association between maternal dietary/supplemental vitamin D intake and maternal serum 25-OHD concentration during each trimester of pregnancy and newborn telomere length measured in cord blood.

## 2. Materials and Methods

### 2.1. Study Participants

We studied data from mothers and infants enrolled in the Maternal Nutrition and Offspring’s Epigenome (MANOE). This is a prospective, observational cohort study initiated in April 2012. Healthy Caucasian women who were in the first trimester of the pregnancy or who desired to become pregnant were recruited at the Department of Obstetrics and Gynecology of the University Hospital Leuven (Belgium). Between April 2012 and May 2018, 214 women were enrolled. 106 mothers-infant pairs were excluded from the study due to the presence of pregnancy complications (gestational diabetes and preeclampsia), preterm birth, birth defect, missing data (no maternal blood or cord blood sample available, no dietary record available). This provided data on a final set of 108 mother–child pairs (Figure 1). Serum 25-OHD was analyzed in 89 women during each trimester of pregnancy. More detailed information about the recruitment process can be found in a previous paper [13].

This study was conducted according to the guidelines laid down in the Declaration of Helsinki and all procedures involving human subjects were approved by the UZ Leuven-Committee for Medical Ethics (ML7975). Written informed consent was obtained from all subjects.

### 2.2. Maternal and Infant Measurements

All 108 women were followed-up during pregnancy at their three scheduled ultrasounds (first trimester (11–13 weeks of pregnancy), second (18–22 weeks of pregnancy), and third trimester (30–34 weeks of pregnancy)) and at delivery. Interviews and questionnaires were used to collect data about socio-demographic factors, lifestyle habits (smoking (yes/no)), and physical activity. To calculate the pre-pregnancy BMI (kg/m^2^), height and pre-pregnancy weight were used. Highest degree of education was stated as a proxy of socio-economic status and was coded as low (degree of high school or 7th year), medium (Bachelor’s and second Bachelor’s degree) and high education (Master’s and second Master’s degree or Ph.D.). A modified version of the Kaiser Physical Activity Survey was used to calculate a physical activity score during each trimester of pregnancy. A total activity index was calculated as the sum of four activity indices (household/caregiving, occupational, active living, sports/exercise). Each activity index has a value that ranged from 1 to 5 representing increasing levels of participation [14]. At birth, information about birth weight, length, and gestational age was obtained from the hospital clinical record and arterial cord blood was drawn. More detailed information about maternal and infant measurements can be found in previous papers [13,15].

### 2.3. Vitamin D

#### 2.3.1. Dietary Vitamin D Intake

A seven-day estimated diet record (7d EDR) was used to calculate the mean daily dietary vitamin D intake of mothers (during each trimester). Guidelines were given for filling in the 7d EDR and a correctly filled in example was provided. They were asked to report all consumed foods and drinks over seven consecutive days. Days were subdivided into six eating occasions: breakfast, morning snacks, lunch, afternoon snacks, dinner, and evening snacks. Detailed information on the type and portion size (expressed as household measures or quantification methods like grams, standard units e.g., a medium-sized apple) of the foods consumed was collected using an open entry format. Only good-quality EDRs were taken into consideration and were encoded and entered in Diet Entry and Storage program NUBEL [16]. In addition, each trimester, questions were asked about the use of nutritional supplements (frequency, brand/type and dosage) to assess the supplemental intake of vitamin D.

#### 2.3.2. Serum 25-hydroxyvitamin D Concentration

Whole blood from mothers during each trimester of pregnancy was collected in 4.5 mL serum tubes. Serum tubes were centrifuged for 10 min at 4000 rpm and serum was stored at −80 °C until analysis. Serum samples were analyzed at the clinical laboratory of UZ Leuven, Belgium. Serum 25-OHD levels were measured using a liquid chromatography tandem-mass spectrometry (LC–MS/MS) (Applied Biosystems, Halle, Belgium) method. Briefly, 50 µL serum and 200 µL precipitation reagent were mixed thoroughly and centrifuged. For precipitation, a mixture of 0.3 M zinc sulphate in water with methanol (20:80 (*v*/*v*)) containing d6-25-OHD3 as an internal standard was used. Methods for 25-hydroxyvitamin D3 and 25-hydroxyvitamin D2 in serum were developed on a 5500 QTRAP mass spectrometer (AB Sciex, Framingham, MA, USA) connected with a (Shimadzu, Kyoto, Japan) Shimadzu chromatographic system. The method consisted of an online cleanup step using a Strata C8 (20 × 2.0 mm, 20 µm, Phenomenex, Torrance, CA, USA) online extraction cartridge. After loading and cleaning, the analytes were eluted by back-flushing. A Kinetex F5 (100 × 3.0 mm, 2.6 µm, Phenomenex) column at 45 °C was used as an analytical column. Atmospheric-pressure chemical ionization (APCI) in positive mode was used as ionization method. Mass transitions of 383.2 > 211.1 and 395.2 > 211.1 were used for 25-hydroxyvitamin D3 and 25-hydroxyvitamin D2 respectively. The method was standardized to the international reference material (National Institute for Standards and Technology (NIST, Gaithersburg, MD, USA). The 3-epi-25-hydroxyvitamin D isobar was fully separated, thus eliminating interference of this compound. Within and between-run imprecision at 28 ng/mL were 4.9% and 6.8% respectively. The limit of quantification was 2.0 ng/mL [17].

### 2.4. Average Relative Telomere Length Measurements

Whole blood samples from the mother and cord blood from the newborn were collected in 4.5 mL tubes containing ethylenediaminetetraacetic acid (EDTA) and were subsequently stored at −20 °C. DNA from whole blood and cord blood was extracted using the salting out method [18]. A NanoDrop 1000 spectrophotometer (Isogen, Life Science, Belgium) was used to determine DNA quantity/purity and DNA samples were stored at −80 °C until analysis [5]. DNA integrity was assessed by agarose gel-electrophoresis and a modified quantitative real-time PCR (qPCR) protocol was used to measure the average relative TL [19,20]. The Quant-iT™ PicoGreen^®^ dsDNA Assay Kit (Life Technologies, Brussels, Belgium, Europe) was used to dilute and check samples to make sure that there was a uniform DNA input of 5 ng for each qPCR reaction. The reaction mixtures that were used for the telomere run, the single copy-gene run, and used PCR cycles can be found in a previous paper [6]. All measurements were performed in triplicate on a 7900HT Fast Real-Time PCR System (Applied Biosystems, Hasselt, Belgium, Europe) in a 384-well format. To assess PCR efficiency, a six-point serial dilution of maternal whole blood or cord blood DNA was used. To account for inter-run variability, five inter-run calibrators (IRC) were used. QBase+2 software (Biogazelle, Zwijnaarde, Belgium) was used to calculate the relative average telomere lengths and they were expressed as the ratio of telomere copy number to single-copy gene number (T/S) relative to the average T/S ratio of the entire sample set (108 maternal and newborn samples). Samples were measured at the Centre for Environmental Sciences, Hasselt University, where an interlaboratory comparison was performed of their telomere assay with a US reference lab to standardize the protocol [5]. Samples for TL were measured in triplicate. The reliability of our assay was assessed by calculating the intraclass coefficient (ICC) with 95% CI of triplicate measures (T/S ratios). Both the inter-assay (IRCs run over multiple qPCR plates) and intra-assay ICC (based on all measures) was calculated using the available online R script on the Telomere Research Network website [21]. The inter-assay ICC was 0.936 (95% CI: 0.808 to 0.969) and the intra-assay ICC was 0.906 (95% CI: 0.885 to 0.922).

### 2.5. Statistical Analysis

All tests were two-sided, a 5% significance level was assumed for all tests. Analyses were performed using SPSS software IBM SPSS Statistics 25 for Windows (NY, USA). Continuous variables were tested for normality. For graphic representations, the GraphPad Prism software version 8 (San Diego, CA, USA) was used. First, we assessed if there is a significant correlation between maternal dietary vitamin D intake/maternal total vitamin D intake (diet + supplements) and maternal serum 25-OHD concentrations using the Pearson correlation. Next, we performed an independent sample t-test to check for significant differences between newborn sex and average relative TL. Next, we assessed the association between newborn TL and maternal vitamin D intake (diet and supplement)/serum 25-OHD concentrations by using generalized linear models during each trimester of pregnancy and the entire pregnancy (mean intake of the 3 trimesters). First, we ran an unadjusted analysis. Second, we adjusted our model for maternal age, education, maternal smoking, pre-pregnancy BMI, newborn sex, and birthweight (model 1). In the next model, additional adjustments were made for season of delivery and mean physical activity during pregnancy (model 2). As newborn TL may be influenced by parental TL, we adjusted for maternal TL in a separate model (model 3). Potential confounders were selected based on the association with TL and the possible association with vitamin D.

## 3. Results

### 3.1. Maternal and Infant Characteristics

Characteristics of the newborns and mothers are presented in Table 1. Of the newborns, there were 59 boys (54.6%) and 49 girls (45.4%). The mean (SD) gestational age was 39.7 ± 0.9 weeks (range: 37.1–41.4). Newborns had a mean birth length of 51.0 ± 1.8 cm (range 47–56) and the mean birth weight was 3512.5 ± 412.9 g (range: 2720–4750). The majority of the newborns were born during spring (*n* = 38; 35.2%).

The mean maternal age was 30.7 ± 3.4 years (range: 24–41), mean pre-pregnancy BMI was 22.9 ± 3.2 kg/m^2^ (range: 17.9–33.0), and mean gestational weight gain was 14.6 ± 4.1 kg (range: 1.9–28.9). Most of the women (51.9%) were highly educated. The physical activity level was highest in the first 12 weeks of pregnancy (10.1 ± 1.3; range: 6.7–13.2) and the lowest in the third trimester (9.5 ± 1.5; range: 5.6–12.7). Three women (2.8%) smoked during pregnancy.

### 3.2. Maternal Vitamin D

Mean maternal dietary vitamin D intake during the entire pregnancy was 3.9 ± 2.6 μg (range: 0.1–14.5). About 50% of the mothers took supplements that contained vitamin D, during the first trimester, 59% during the second trimester, and 65% during the third trimester. Mean supplement vitamin D intake during the entire pregnancy was (6.1 ± 4.4 μg; range 0–10). Total (diet + supplement) mean vitamin D intake during the entire pregnancy was 8.9 ± 5.5 μg (range: 0.2–22.3). Maternal serum 25-OHD concentrations were highest in the third trimester of pregnancy (24.8 ± 9.6 ng/mL; range: 8.1–58.4). Based on the serum 25-OHD levels, 36% of the women in our study had a vitamin D deficiency (<20 ng/mL) and 46% insufficiency (21–29 ng/mL). Correlations between maternal dietary vitamin D intake and serum 25-OHD concentrations were not significant (r: 0.047–0.078). However, correlations between total vitamin D intake (diet and supplement) and serum 25-OHD concentrations were significant at each trimester and the mean during the entire pregnancy (r: 0.314–0.573). (Table 2)

### 3.3. Offspring Telomere Length

The mean average relative TL was lower for boys (1.29 ± 0.23, range 0.78–1.79) than for girls (1.39 ± 0.19, range 0.91–1.71) (*p*-value = 0.012). Maternal TL was significantly correlated with newborn TL (r = 0.441, *p*-value < 0.001).

### 3.4. Maternal Vitamin D and Average Relative Newborn TL

Maternal total vitamin D intake (diet + supplement) during the first trimester of pregnancy was positively associated with the average relative TL of the newborn (Table 3). This was observed in both an unadjusted model (*p* = 0.036) and after adjustment for maternal age, maternal education and smoking status, pre-pregnancy BMI, newborn sex, and birthweight (*p* = 0.005). Additionally, adjusting for season of delivery and mean physical activity during pregnancy did not alter this association (*p* = 0.007). Finally, in a supplementary model, we adjusted for maternal average relative TL. We did this to rule out if the association between vitamin D intake and newborn TL is originating from the impact of vitamin D intake on the maternal average relative TL, but the association remained significant (*p* = 0.001).

A negative association was observed between maternal dietary vitamin D intake during the second trimester of pregnancy and newborn TL, only after adjustment (model 1, *p* = 0.028; model 2, *p* = 0.044). (Table 4)

For maternal supplemental intake of vitamin D, positive associations were found with newborn TL during the first trimester of pregnancy and the whole pregnancy (Table 5). Finally, only in models adjusting for maternal TL, the association between maternal total vitamin D and newborn TL tended to be negative (*p* = 0.065), and this was also observed when studying dietary vitamin D (*p* = 0.088) and supplemental vitamin D (*p* = 0.038) separately.

No associations were found between maternal serum 25-OHD concentrations and the average relative TL of the newborn. (Table 6)

## 4. Discussion

In this study, we evaluated the association between maternal vitamin D status and newborn TL. Maternal vitamin D status was studied by extensive dietary questionnaires, vitamin D supplement questionnaires, and serum evaluation of 25-OHD during each trimester of pregnancy. Our results support to some extent the hypothesis that maternal nutrition during pregnancy (first trimester) is associated with telomere length at birth. When evaluating the association between total maternal vitamin D intake (diet + supplement) and average relative TL of the newborn, we found a positive association in the first trimester of pregnancy. Given that effects of vitamin D are more pronounced during the earlier gestational period [22], our results of dietary vitamin D intake are rather remarkable. However, against our expectations, we did not observe any association between maternal serum 25-OHD concentrations and the average relative TL of the newborn during the different stages of pregnancy.

To date, only a few studies have focused on early stages of pregnancy and vitamin D status—both dietary and serum 25-OHD [22]. The women in our study had a mean dietary vitamin D intake during the entire pregnancy of 3.9 μg per day, ranging from 0.1–14.5. This is in line with the results from the Belgium National Food Consumption Survey, where the mean vitamin D intake of adult women (18 and 39 years) was 3.42 μg/day [23]. In our study, the majority of the pregnant women took supplements that contained vitamin D (50% during the first trimester, 59% during the second trimester, and 65% during the third trimester). The mean vitamin D intake through supplements was 8.73 μg/day and ranged from 1.1–10 μg/day. This is insufficient compared with guidelines of the Belgian Superior Health Council for supplementation of pregnant women, where it is advised to supplement with 20 μg vitamin D/day [24]. When nutritional and supplemental vitamin D intake was combined, a mean intake of 8.89 μg/day was found, ranging from 0.2–22.3 μg, which is in line with the mean vitamin D intake (8.10 μg/day) for non-pregnant women in Belgium [23]. The mean serum 25-OHD level during pregnancy was 23.36 ng/mL, ranging from 10.6–45.5 ng/mL [25]. In line with our results, several other studies have observed lower levels of serum 25-OHD in the first trimester as compared to the latter two [26,27]. This may be due to the significantly higher conversion rate of 25-OHD to 1.25(OH)D during the first trimester, as compared to the conversion of vitamin D to 25-OHD, which remains unchanged during the same period [28]. This can potentially be linked to the increase in progesterone levels over the course of pregnancy, which inhibits the 24-hydroxylase (CYP24A1) thereby reducing the breakdown of 25-OHD. While higher levels of 25-OHD as a result of Vitamin D supplementation have been observed in a Swiss cohort [22], vitamin D insufficiency has been observed in British [29] and American [30] cohorts despite supplementation. It has been suggested that maternal characteristics (baseline 25-OHD, age, weight, weight gain during pregnancy) can have an influence on 25-OHD status [29]. Based on these results, we can say that vitamin D intake through diet/supplements and serum 25-OHD concentrations in Belgian pregnant women needs to be improved. Hollis et al. [31] suggest that a circulating concentration of 40 ng/ML (mean of 23.36 ng/mL in our study) should be maintained throughout pregnancy. Higher 25-OHD concentrations can be obtained by improving the intake of vitamin D rich foods (oily fish), eating foods fortified with vitamin D (milk), and vitamin D supplementation (Hollis et al. [31] suggest a supplementation with 100 µg/d) [2].

While a clear mechanistic insight into the regulation of newborn TL by maternal vitamin D levels is not possible from the present study, it can be speculated that 25-OHD-mediated regulation of hypoxia-inducible factor 1, during the hypoxic early pregnancy stage, leads to upregulation of hTERT (and telomerase activity) [32,33].

For the newborns, we found that the mean average relative TL was significantly shorter for the boys compared to that of the girls, as consistent with recent observations [8,34]. Estimates suggest a high heritability in TL (34–80%) and dynamics, and such heritability can often be explained by a combination of genetic and environmental factors.

Our study has several strengths. First, the MANOE birth cohort allowed us to collect longitudinal data of pregnant women. Women were followed-up during the first, second, and third trimester. At each time point, we calculated vitamin D intake through the use of dietary records and supplement use. Next to the dietary intake and supplemental intake of vitamin D during each trimester of pregnancy, we collected a blood sample at each time point to analyze maternal 25-OHD concentrations in serum by using LC-MS/MS, which is considered the gold standard [35]. In addition, we have detailed covariate data allowing for adjustment for potential confounding variables. Furthermore, we were able to control our models for maternal TL. We address the following limitations. First, we were not able to control our models for differential blood cell counts, as this was not available, and TL may be cell type-dependent. Second, the generalizability of our study to the population at large may be limited as the MANOE cohort contains a homogenous group of participants from only Caucasian women and without pregnancy complication, which were excluded. Third, we acknowledge that our study sample is rather small, and that our results are rather exploratory, therefore needing further validation and confirmation in a larger study population. A last limitation is the missing information of the women’s sun exposure during pregnancy, as sun exposure is a main determinant of 25-OHD.

## 5. Conclusions

This study shows that maternal vitamin D intake through diet and supplement, specifically during the first trimester of pregnancy, may be an important factor associated with TL at birth. Further studies, using a larger sample size, are needed to further explore these findings.

## Figures and Tables

**Figure 1 nutrients-13-02012-f001:**
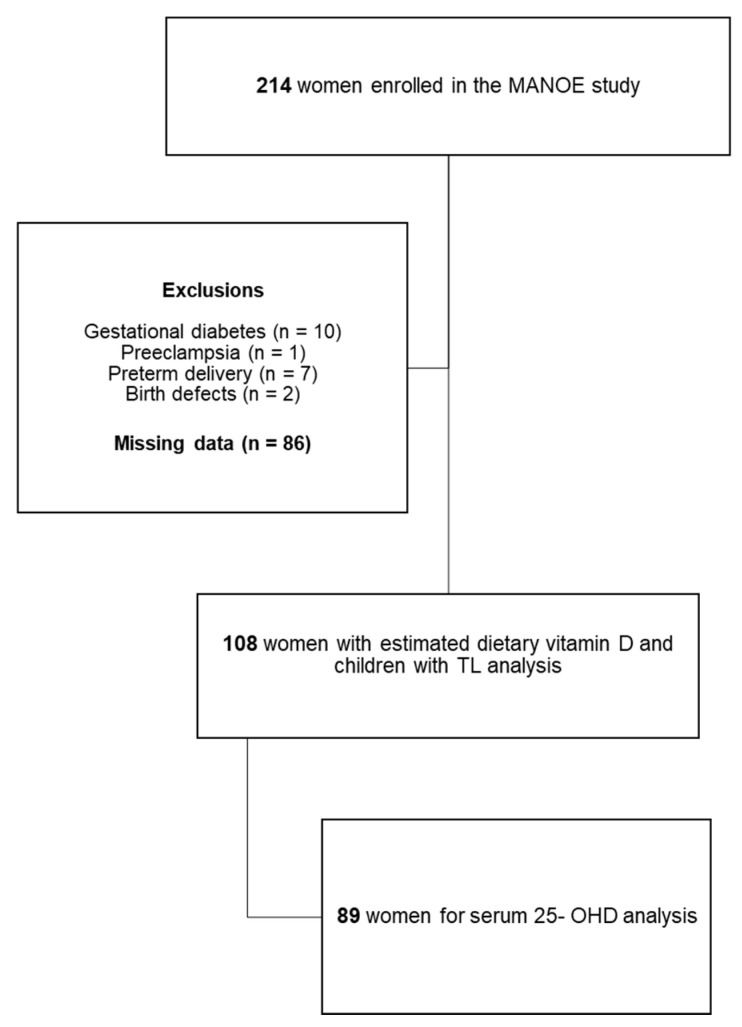
Flowchart of mothers and children enrolled in the MANOE study.

**Table 1 nutrients-13-02012-t001:** Characteristics of the newborns and mothers.

Newborn (*n* = 108)	Mean (SD)	Range
Gestational age, weeks	39.7 (0.9)	37.1–41.4
Length, cm	51.0 (1.8)	47–56
Birth weight, g	3512.5 (412.9)	2720–4750
	*n*	%
Gender		
Boys	59	54.6
Girls	49	45.4
Season of birth		
Winter	21	19.4
Spring	38	35.2
Summer	23	21.3
Autumn	26	24.1
Mother (*n* = 108)	Mean (SD)	Range
Age, years	30.7 (3.4)	24–41
Pre-pregnancy BMI, kg/m^2^	22.9 (3.2)	17.9–33.0
Gestational weight gain, kg	14.6 (4.1)	1.9–28.9
Physical activity index		
First trimester	10.1 (1.3)	6.7–13.2
Second trimester	9.8 (1.5)	6.4–13.7
Third trimester	9.5 (1.5)	5.6–12.7
	*n*	%
Education		
Low education	14	13.0
Medium education	38	35.2
High education	56	51.9
Smoking		
First trimester	3	2.8
Second trimester	2	1.9
Third trimester	2	1.9

**Table 2 nutrients-13-02012-t002:** Maternal vitamin D intake through diet and supplements and serum 25-OHD status during pregnancy.

Maternal	Diet (μg)		Supplements (μg)		Total (μg)		Serum 25-OHD (ng/mL)		Correlation *			
Vitamin D	Mean (SD)	Range	Mean (SD)	Range	(Diet + Supplement)		Mean (SD)	Range	Diet/Serum		Total/Serum	
					Mean (SD)	Range			R	*p*	R	*p*
Trimester 1	3.8 (2.3)	0.2–10	4.4 (4.8)	0–10	8.2 (5.4)	0.2–17.9	22.4 (8.1)	6.5–68	0.078	0.439	0.314	**0.001**
Trimester 2	3.9 (2.8)	0.5–14.5	5.3 (4.8)	0–10	9.3 (5.6)	0.7–22.3	23.0 (8.1)	7.7–52	0.056	0.575	0.535	**<0.001**
Trimester 3	3.9 (2.5)	0.1–10.7	5.4 (4.6)	0–10	9.3 (5.5)	0.4–20	24.8 (9.6)	8.1–58.4	0.063	0.535	0.468	**<0.001**
Entire pregnancy	3.9 (2.6)	0.1–14.5	6.1 (4.4)	0–10	8.9 (5.5)	0.2–22.3	23.4 (6.8)	10.6–45.5	0.047	0.656	0.573	**<0.001**

* Correlation presented as Pearson correlation test with corresponding *p*-value.

**Table 3 nutrients-13-02012-t003:** Maternal total vitamin D intake (diet + supplement) during the first, second, and third trimester and entire pregnancy and average relative telomere length (TL) of the newborn.

	Entire Pregnancy		First Trimester		Second Trimester		Third Trimester	
	β (95% CI)	*p*-Value	β (95% CI)	*p*-Value	β (95% CI)	*p*-Value	β (95% CI)	*p*-Value
**Unadjusted model**	0.004 (−0.005, 0.013)	0.365	0.009 (0.001, 0.018)	**0.036**	−0.003 (−0.013, 0.007)	0.534	−0.001 (−0.010, 0.008)	0.820
**Model 1**	0.007 (−0.002, 0.015)	0.132	0.012 (0.004, 0.020)	**0.005**	−0.004 (−0.013, 0.006)	0.444	−0.001 (−0.009, 0.008)	0.828
**Model 2**	0.007 (−0.002, 0.016)	0.121	0.012 (0.003, 0.020)	**0.007**	−0.003 (−0.012, 0.007)	0.569	−0.001 (−0.010, 0.007)	0.763
**Model 3**	0.003 (−0.005, 0.010)	0.498	0.012 (0.005, 0.019)	**0.001**	−0.002 (−0.010, 0.006)	0.569	−0.007 (−0.014, 0.000)	0.065

Unadjusted model: Model 1: adjustment for maternal age, maternal education and smoking status, pre-pregnancy BMI, gender of the newborn and birth weight; Model 2: additional adjustment for season of delivery and mean physical activity during pregnancy; Model 3: additional adjustment for maternal average relative TL.

**Table 4 nutrients-13-02012-t004:** Maternal dietary vitamin D intake during the first, second, and third trimester and entire pregnancy and average relative TL of the newborn.

	Entire Pregnancy		First Trimester		Second Trimester		Third Trimester	
	β (95% CI)	*p*-Value	β (95% CI)	*p*-Value	β (95% CI)	*p*-Value	β (95% CI)	*p*-Value
**Unadjusted model**	−0.017 (−0.041, 0.006)	0.148	0.002 (−0.016, 0.020)	0.832	−0.011 (−0.026, 0.004)	0.162	−0.005 (−0.022, 0.012)	0.542
**Model 1**	−0.014 (−0.037, 0.009)	0.237	0.011 (−0.006, 0.029)	0.202	−0.017 (−0.032, −0.002)	**0.028**	−0.002 (−0.018, 0.013)	0.782
**Model 2**	−0.019 (−0.043, 0.005)	0.124	0.009 (−0.010, 0.028)	0.345	−0.015 (−0.030, 0.000)	**0.044**	−0.005 (−0.021, 0.011)	0.510
**Model 3**	−0.018 (−0.038, 0.002)	0.077	0.005 (−0.011, 0.020)	0.547	−0.008 (−0.021, 0.005)	0.213	−0.012 (−0.025, 0.002)	0.088

Unadjusted model: Model 1: adjustment for maternal age, maternal education and smoking status, pre-pregnancy BMI, gender of the newborn and birth weight; Model 2: additional adjustment for season of delivery and mean physical activity during pregnancy; Model 3: additional adjustment for maternal average relative TL.

**Table 5 nutrients-13-02012-t005:** Maternal supplemental vitamin D intake during the first, second, and third trimester and entire pregnancy and average relative TL of the newborn.

	Entire Pregnancy		First Trimester		Second Trimester		Third Trimester	
	β (95% CI)	*p*-Value	β (95% CI)	*p*-Value	β (95% CI)	*p*-Value	β (95% CI)	*p*-Value
**Unadjusted model**	0.009 (−0.0011, 0.018)	**0.050**	0.010 (0.000, 0.020)	0.062	0.002 (−0.012, 0.015)	0.881	−0.003 (−0.016, 0.010)	0.654
**Model 1**	0.010 (0.001, 0.018)	**0.024**	0.010 (0.001, 0.020)	**0.039**	0.005 (−0.008, 0.018)	0.454	−0.006 (−0.018, 0.007)	0.367
**Model 2**	0.010 (0.001, 0.019)	**0.028**	0.010 (−0.002, 0.020)	**0.050**	0.006 (−0.007, 0.019)	0.388	−0.005 (−0.017, 0.007)	0.413
**Model 3**	0.005 (−0.003, 0.012)	0.250	0.012 (0.004, 0.021)	**0.005**	0.003 (−0.008, 0.014)	0.552	−0.011 (−0.021, −0.001)	**0.038**

Unadjusted model: Model 1: adjustment for maternal age, maternal education and smoking status, pre-pregnancy BMI, gender of the newborn and birth weight; Model 2: additional adjustment for season of delivery and mean physical activity during pregnancy; Model 3: additional adjustment for maternal average relative TL.

**Table 6 nutrients-13-02012-t006:** Maternal serum 25-OHD concentration during the first, second, and third trimester and entire pregnancy and average relative TL of the newborn.

	Entire Pregnancy		First Trimester		Second Trimester		Third Trimester	
	β (95% CI)	*p*-Value	β (95% CI)	*p*-Value	β (95% CI)	*p*-Value	β (95% CI)	*p*-Value
**Unadjusted model**	0.004 (−0.003, 0.011)	0.252	0.003 (−0.005, 0.012)	0.466	−0.003 (−0.013, 0.006)	0.500	0.005 (−0.002, 0.011)	0.166
**Model 1**	0.003 (−0.003, 0.011)	0.369	0.004 (0.004, 0.013)	0.324	−0.004 (−0.013, 0.005)	0.422	0.004 (−0.003, 0.011)	0.284
**Model 2**	0.004 (−0.003, 0.010)	0.306	0.003 (−0.006, 0.013)	0.496	−0.002 (−0.012, 0.009)	0.716	0.003 (−0.006, 0.011)	0.511
**Model 3**	0.003 (−0.003, 0.009)	0.311	0.003 (0.005, 0.011)	0.517	0.000 (−0.009, 0.009)	0.962	0.000 (−0.007, 0.008)	0.934

Unadjusted model: Model 1: adjustment for maternal age, maternal education and smoking status, pre-pregnancy BMI, gender of the newborn and birth weight; Model 2: additional adjustment for season of delivery and mean physical activity during pregnancy; Model 3: additional adjustment for maternal average relative TL.

## Data Availability

The data presented in this study are available on request from the corresponding author. The data are not publicly available due to privacy/ethical restrictions.

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
