# Peer review of "Maternal Vitamin D and Newborn Telomere Length"

_nutrients, 2021, doi:10.3390/nu13062012_

Round 1
Reviewer 1 Report
This is a unique study that presents conflicting results. On one hand vitamin D intake apparently can affect telomere length but this cannot be related to circulating 25D levels. It would have been helpful to measure circulating 1,25D levels in these subjects as that metabolite in pregnancy is very important. In fact the authors should read, cite and discuss the RCT of vitamin D during pregnancy published by Hollis et al 2011 JBMR. Overall, the levels of circulating 25D in these subjects are very low as is reflected by their overall poor vitamin D intake and the need to increase this should be discussed.
Reviewer 2 Report
This is a clear, concise and well written manuscript on the associations of maternal vitamin D status and vitamin D intakes (diet, supplemental and total) and newborn telomere length. Please see my specific comments below.
Throughout manuscript:
- It is conventional to describe 25(OH)D status in either nmol/L or ng/mL (as opposed to µg/L). To improve the readability of the manuscript, I would suggest amending the 25(OH)D status to either nmol/L or ng/mL.
Abstract:
- Line 23, change ‘participants’ to ‘women’.
Introduction:
- Line 38, delete “etc.” and list more fully the factors contributing to low vitamin D status in Europe.
- Lines 58-59, use 25(OH)D abbreviation as you have previously defined this abbreviation in line 37.
Methods:
- Line 65, in the sub-title, change ‘study subjects’ to ‘study participants’.
- I could not find any reference to the trial registration. Was the trial registered (e.g. at clinicaltrials.gov)? If so, this should be mentioned in the Methods section.
- Was data on women’s sun exposure during pregnancy trimesters collected? If not this should be recognized as a limitation of the study.
- Line 176, it was not clear to me how entire pregnancy vitamin D intake and 25(OH)D was assessed. The authors should make this more clear in the statistical analysis section.
Results:
- The title for Table 2 is very poor and seems incomplete? This table title should be expanded for completeness.
- Delete Table 3. This is just a repeat of the text description above (lines 216-218) and is not adding any further information (add ranges to text). It is redundant and should therefore be removed.
Discussion:
- Lines 316-317, “Given that effects of vitamin D are more pronounced during the earlier gestational period,…..” add a reference.
- Lines 323-324, two vitamin D intakes are given but it is not clear what these relate to as the first part of the sentence only refers to the mean vitamin D intake of the women in the study. Please clarify.
- Line 331, which guidelines are the authors referring to when they state it is advised that pregnant women need to take 20 µg/d vitamin D supplement during pregnancy? Are these Belgian national recommendations? Recommendations by SACN for the UK, EFSA for Europe and the IOM for North America, state 10, 15 and 15 µg/d respectively for pregnant women. Therefore, it is not clear to me which recommendations these guidelines are referring to. Please add a reference.
- Lines 350-353, the study by Kim et al. did not assess maternal vitamin D intakes with telomere length, only maternal 25(OH)D status. Therefore, in line 352, the reference to vitamin D intakes in the Kim et al. study needs to be deleted. As no associations were found between matenal 25(OH)D concentrations and newborn telomere length in this study, this sentence needs to be amended to reflect this in comparison to the Kim et al. study.
- A strength of the study is the gold standard assessment of 25(OH)D by LC-MS/MS. As clearly explained by Sempos and Binkley (Public Health Nutrition 2020; 23(7): 1153-1164), standardization of vitamin D assays are needed if we are to provide evidence to underpin updated vitamin D guidelines.
Conclusions:
- Line 374, delete the word ‘in-take’ after supplement.
